# Biostimulatory Effects of Foliar Application of Silicon and *Sargassum muticum* Extracts on Sesame Under Drought Stress Conditions

**DOI:** 10.3390/plants14152358

**Published:** 2025-07-31

**Authors:** Soukaina Lahmaoui, Rabaa Hidri, Hamid Msaad, Omar Farssi, Nadia Lamsaadi, Ahmed El Moukhtari, Walid Zorrig, Mohamed Farissi

**Affiliations:** 1Laboratory of Biotechnology & Sustainable Development of Natural Resources, Polydisciplinary Faculty, Sultan Moulay Slimane University, P.O. Box 592, Beni Mellal 23000, Morocco; soukaina.lahmaoui@usms.ac.ma (S.L.); hamid.msaad@usms.ac.ma (H.M.); omarfarssi@gmail.com (O.F.); 2Laboratory of Extremophile Plants, Centre of Biotechnology of Borj-Cedria, P.O. Box 901, Hammam-Lif 2050, Tunisia; hidrirabaa@gmail.com (R.H.); zorrigwalid@gmail.com (W.Z.); 3Laboratory of Scientific Innovation in Sustainability, Environment, Education, and Health in the Era of Artificial Intelligence, Higher School Teachers-Training Institution, Mohamed Ben Abdellah University, Fes 30000, Morocco; lamsaadinadia2015@gmail.com; 4Laboratory of Ecology and Environment, Faculty of Sciences Ben M’Sick, Hassan II University of Casablanca, Casablanca 20360, Morocco; elmoukhtari.ahmed94@gmail.com

**Keywords:** stomatal conductance, quantum yield, seaweed extract, silicon, photosynthetic parameters, photosystem I, photosystem II, biomass production

## Abstract

Sesame (*Sesamum indicum* L.) is widely cultivated for its valuable medicinal, aromatic, and oil-rich seeds. However, drought stress remains one of the most significant abiotic factors influencing its development, physiological function, and overall output. This study investigates the potential of foliar applications of silicon (Si), *Sargassum muticum* (Yendo) Fensholt extracts (SWE), and their combination to enhance drought tolerance and mitigate stress-induced damage in sesame. Plants were grown under well-watered conditions (80% field capacity, FC) versus 40% FC (drought conditions) and were treated with foliar applications of 1 mM Si, 10% SWE, or both. The results showed that the majority of the tested parameters were significantly (*p* < 0.05) lowered by drought stress. However, the combined application of Si and SWE significantly (*p* < 0.05) enhanced plant performance under drought stress, leading to improved growth, biomass accumulation, water status, and physiological traits. Gas exchange, photosynthetic pigment content, and photosystem activity (PSI and PSII) all increased significantly when SWE were given alone; PSII was more significantly affected. In contrast, Si alone had a more pronounced impact on PSI activity. These findings suggest that Si and SWE, applied individually or in combination, can effectively alleviate drought stress’s negative impact on sesame, supporting their use as promising biostimulants for enhancing drought tolerance.

## 1. Introduction

Ancient civilizations used medicinal plants for healing. Many studies recognize their active role in pharmacotherapy, enhancing pharmacists’ ability to address challenges and improve human life [1]. Recently, abiotic stressors have increased as a result of climate change; one such abiotic stress that significantly reduces agricultural yield annually is drought stress [2]. Drought in agriculture is worsened by water scarcity and further exacerbated by rising food demand due to global population expansion, which places a lot of pressure on agriculture globally [3].

The effects of drought stress on the development and metabolism of plants are caused by interrelated physiological and biochemical processes. Plants have less access to water when soil moisture levels drop [4], which lowers cell turgor and structural integrity and can result in leaf wilting and slower growth. One of the primary responses to drought stress is stomatal closure; it restricts photosynthetic activity by limiting water loss through transpiration while also lowering CO_2_ uptake [5]. As a result of variations in sugar concentrations, this restriction in CO_2_ availability affects cellular osmotic potential, disrupts carbon fixation, and lowers glucose and energy generation for plant metabolism [6].

Additionally, stomatal closure promotes the accumulation of reactive oxygen species (ROS), such as hydrogen peroxide (H_2_O_2_) and superoxide (O_2_^−^), which impair photosynthetic metabolism and further restrict CO_2_ absorption by leaves [7,8]. To mitigate these effects, plants activate various adaptive mechanisms, including the accumulation of compatible solutes, such as sugar, proline, and polyols, to maintain osmotic balance, stabilize proteins, and protect cellular membranes. Furthermore, drought stress triggers hormonal signaling pathways, particularly involving abscisic acid (ABA), which plays a crucial role in stress response regulation by inducing stomatal closure, activating stress-related genes, and modulating other metabolic processes [9]. In addition, a notable decrease in the photosynthetic electron transport chain during drought can indirectly impact the total photosynthetic rate [10,11]. These physiological alterations collectively lead to reduced growth rates, biomass accumulation, and changes in water and nutrient uptake dynamics, ultimately compromising plant productivity and water-use efficiency [12,13,14,15]. Plants also modify protein synthesis and increase antioxidant production to reduce oxidative stress, which contributes to improved drought tolerance.

*Sesamum indicum* L. (Sesame) is known to be one of the oldest oilseed crops and is widely cultivated for the historical uses of its seeds, seed oil, and plant organs, which have been used to cure ailments such as ulcers, asthma, wounds, amenorrhea, hemorrhoids, and inflammation [16]; it also has culinary uses and is used in cosmetics. Crops of medicinal and aromatic plants, such as sesame, are also affected by drought throughout their different development stages, including seed germination, seedling establishment, root and shoot growth, blooming, and pollination, and are highly vulnerable to dryness [17]. In Morocco, the Beni-Mellal/Khenifra region alone provides 90% of total sesame production [18], making it an interesting plant to focus on for its sustainable production. Identifying strategies to mitigate water deficit stress in plants has become a must in recent years seeing that droughts occur regularly; this has necessitated the development of strategies aimed at reducing the impacts of drought [19] and developing drought-tolerant crops [20].

Hence, the need for more sustainable and efficient strategies is always a priority in searching for long-term solutions, and the use of biostimulants based on natural resources such as seaweeds [21] and silicon [22] could provide an alternative solution for ensuring resilient agriculture in the context of climate change.

Silicon (Si) is the second most abundant element in the Earth’s crust [23]. Cao et al. (2017) [24] found that silicon can be applied using different methods, such as foliar spraying, direct soil application, hydroponic systems, or seed priming. Silicon application is an environmentally friendly approach to enhancing plant responses to abiotic stresses [25,26]. Si effectively reduces water stress effects by improving gas exchange, water transportation, leaf water potential, and chlorophyll content [27,28,29].

Si plays a crucial role in protecting and enhancing the physiological and biochemical performance of plants, particularly under stress conditions [30]. It preserves photosynthetic pigments, such as chlorophylls, by reducing their degradation, thereby improving photosynthetic efficiency and light energy capture [31,32]. Si also stimulates the activity of photosystems I and II (PSI and PSII), increasing quantum yield and reducing oxidative damage through a decrease in ROS [33]. Furthermore, silicon strengthens water and nutrient absorption, promoting the better hydration of tissues, as observed by an increase in relative water content [8]. Finally, it enhances both fresh and dry biomass of plants by optimizing stomatal regulation, cellular structure, and activating antioxidant systems, thereby providing improved stress tolerance and promoting increased growth [34,35].

In addition to Si, another promising natural resource for enhancing plant performance under stress is seaweed extracts (SWE), which also offer numerous benefits for plant growth and resilience.

While SWE are rich in nutrients, plant hormones, and bioactive substances, they are essential for promoting plant growth and development [36]. They contain auxins and cytokinins that promote root and shoot growth, thereby enhancing water and nutrient absorption [37]. Furthermore, their application significantly increases plant height and leaf numbers, indicating enhanced vegetative growth and a larger photosynthetic surface area [38]. By improving the efficiency of photosystem I and II, these extracts optimize light absorption and chlorophyll production, facilitating gas exchange and CO_2_ uptake while minimizing water loss [39]. Additionally, the polysaccharides present in seaweed extracts contribute to soil structure improvement, facilitating nutrient availability for root systems [40]. By stimulating beneficial metabolic responses, seaweed extracts also strengthen plant resistance to environmental stresses [41,42]. Collectively, these effects position seaweed extracts as a natural and effective approach to optimizing crop health and productivity. SWE are also beneficial for sustainable agriculture, since they enhance leaf health and increase fruit yield and quality [43].

The biostimulant effects of seaweed extracts and silicon have been well documented in various crops, especially for improving stress tolerance and physiological performance. However, their combined application has not yet been investigated in sesame (*Sesamum indicum* L.), particularly under drought conditions. Therefore, the primary objective of this study is to evaluate the individual and combined effects of the seaweed extract of *Sargassum muticum* (Yendo) Fensholt and silicon as biostimulants to alleviate drought stress in sesame. Specifically, this study aims to determine whether Si and SW treatments enhance sesame growth and physiological traits under limited water availability, assess their influence on photosynthetic efficiency, focusing on the performance of photosystems I and II (PSI and PSII), and clarify any synergistic effects when both biostimulants are applied together. We hypothesize that these biostimulants mitigate drought stress primarily by improving photosynthetic processes and maintaining better physiological function in sesame.

## 2. Results

### 2.1. Morphological Indicators of Growth

Sesame plants exhibited a reduction in size when subjected to water stress, along with visible chlorosis compared to the control group. Compared to the well-watered control, drought stress caused reductions across all morphological parameters, confirming the sensitivity of sesame to water deficit. However, foliar spray treatments with Si, SWE alone, or their combination resulted in general improvements in plant size and morphology. Notably, the combined treatment (Si + SWE) showed the most significant enhancement in morphology, both under stress conditions and in the absence of stress (Figure 1a–c).

Biostimulant treatments significantly improved growth parameters, not only under drought conditions, but also under optimal irrigation. Under stress, Si alone increased shoot length (SL) by 26% (Figure 1d) and root length (RL) by 34% (Figure 1e) compared to drought-stressed controls, while SWE alone enhanced SL by 52% and RL by 57%. The combined treatment of Si and SWE boosted SL by 55% and RL by 64%.

On the other hand, water stress significantly reduced SD in sesame (Figure 2a). Compared to untreated controls, this parameter decreased by 41% under water stress. However, the supplementation of Si, SWE or their combination to drought-stressed plants improved SD by 31%, 18%, and 48%, respectively. Under well-watered conditions, the combined treatment showed the highest values among all treatments (*p* < 0.05).

For dry biomass, drought stress caused a significant (*p* < 0.05) reduction in SDW, while the effect on RDW was statistically insignificant, with reductions of 55% and 81%, respectively. The whole plant dry weight (WPDW) decreased by 58% due to water deficit. As shown in Figure 2b–d, treatment with Si resulted in increases in RDW, SDW, and WPDW relative to drought-stressed plants. SWE alone produced similar effects. Importantly, the combined treatment to the stressed plants increased RDW, SDW, and WPDW by 93%, 80%, and 82%, respectively. Under optimal conditions, the combined application also led to a slight enhancement of dry biomass, suggesting a general growth promoting effect independent of stress.

### 2.2. Number of Leaves and Leaf Area

Sesame plants subjected to water stress exhibited a marked reduction in both leaf area (LA) (Figure 3b) and the number of leaves (NL) (Figure 3a), with decreases of 64% and 24%, respectively, compared with well-watered controls (Figure 3). The application of Si to water-stressed plants significantly enhanced LA by 126% and the NL by 21%. A similar beneficial effect was observed in plants treated with SWE, which raised these parameters by nearly 97% and 24%, respectively. The combined Si + SWE treatment improved LA by 224% and LN by 31% under stress conditions. Nevertheless, these improvements did not fully restore leaf characteristics to those of well-watered plants, indicating that a 50% reduction in irrigation constitutes a highly severe stress for sesame.

### 2.3. Leaf Relative Water Content (LRWC) and Leaf Water Potential (LWP)

Drought stress elicited a 24% reduction in LRWC compared with the well-watered control (Figure 4a), demonstrating that a 50% irrigation deficit markedly compromises sesame’s capacity to maintain foliar hydration. However, through the application of Si, SWE, and their combination, plants significantly ameliorated LRWC by 19%, 87%, and 32%, respectively. Under drought stress, LWP became significantly more negative (110% compared to the control; *p* < 0.05), reflecting a marked decline in leaf water status (Figure 4b). Treatments with Si, SWE, and Si + SWE moderated this effect by 59%, 55%, and 70%, respectively, relative to the stressed control.

### 2.4. Photosynthetic Pigment Content

Drought stress significantly reduced chlorophyll *a* (chl. *a*), chlorophyll *b* (chl. *b*), total chlorophyll (chl. *a* + *b*), and carotenoids (car.) contents by 34%, 37%, 35%, and 31%, respectively (Figure 5a–d). However, Si treatment alleviated the water stress effect on chl. *a*, chl. *b*, total chl., and car. by 20%, 54%, 47%, and 35%, respectively. In the same sense, SWE alleviated the effect on the same parameters by 62%, 78%, 65%, and 49%. The combined Si + SWE treatment was generally less effective, giving intermediate values. In well-watered plants, Si and SWE increased the pigment parameters by 23%, 25%, 23%, and 21%, respectively, for Si and by 14%, 22%, 16%, and 19% for SWE, whereas the Si + SWE combination produced smaller improvements. Soil–Plant Analysis Development (SPAD) values followed the same pattern: drought stress led to a significant decline, while all treatments improved chl. contents (Figure 5e). Nonetheless, pigment levels under treatment remained below those of well-watered plants, confirming that the 50% irrigation deficit remains a strong stress factor.

### 2.5. Photosynthetic Gas Exchange

Sesame plants exposed to drought stress displayed markedly lower gas exchange performance, with transpiration rate (*E*), stomatal conductance (*gs*) and net CO_2_ assimilation (*A*) declining by 62%, 53%, and 79%, respectively, relative to well-watered controls (Figure 6a–c). Foliar application of Si or SWE under water stress significantly improved all three parameters. In fact, Si increased *E*, *gs*, and *A* by 64%, 89%, and 208%, whereas SWE boosted them by 222%, 318%, and 362%, respectively. A combined Si + SWE spray enhanced the *E*, *gs*, and *A* by 192%, 328%, and 293% over the stressed control. Under full irrigation, the tested biostimulants also enhanced gas exchange traits, with SWE alone raising *E*, *gs*, and *A* by 17%, 47%, and 51%, whereas its combination with the Si + increases all parameters by 46%, 89%, and 208% relative to the untreated control. In every case, SWE outperformed Si when the two were applied separately, while the combined treatment delivered the broadest overall stimulation. Nonetheless, even the best drought side treatments did not fully restore gas exchange rates to those observed in well-watered plants, again highlighting that a 50% reduction in irrigation constitutes an excessively severe stress for sesame (Figure 6).

### 2.6. Activities of PSI and PSII

Sesame plants exposed to drought stress exhibited a marked decline in chlorophyll fluorescence parameters of dark-adapted leaves. The maximal quantum yield of PSII (Fv/Fm) decreased by 31% compared to the well-watered control (Figure 7a–c).

Foliar treatment of drought-stressed plants with Si, SWE, or their combination improved Fv/Fm by 53%, 44%, and 49%, respectively, compared to drought-stressed plants without treatment, although values remained below those observed in fully irrigated controls.

A similar trend was observed for P700 parameters. Under drought stress, P700ox and P700m decreased by 35% and 18%, respectively. Biostimulation with Si increased P700ox and P700m by 69% and 131%; SWE by 39% and 131%; and Si + SWE by 9% and 87%, respectively (Figure 7).

Under well-watered conditions, application of Si and SWE led to only modest improvements in Fv/Fm, P700ox, and P700m, confirming that biostimulants primarily benefit plants under drought stress but do not exceed the performance of non-stressed controls.

### 2.7. Quantum Yield and Energy Conversion in Photosystem II

The results demonstrated that the quantum yields of photochemical and non-photochemical energy conversion in PSII; namely, the effective quantum yield of PSII photochemistry (Y(II)), the regulated non-photochemical energy dissipation (Y(NPQ)), and the electron transport rate in PSII (ETR (II)) were significantly affected by drought stress (Figure 8a,b,d). Sesame plants grown under drought stress conditions exhibited a gradual decrease in Y(II), Y(NPQ), and ETR (II), with increasing photosynthetically active radiation (PAR), compared to the control. Interestingly, drought-stressed plants sprayed with Si and SWE, either alone or in combination, restored the control levels of Y(II), Y(NPQ), and ETR(II).

In contrast, the quantum yield of non-regulated energy dissipation in PSII (Y(NO)) of drought-stressed plants showed a markedly higher curve (Figure 8c), indicative of increased non-regulated energy loss; this was also mitigated by Si and SWE supplementation, bringing values back to more favorable levels.

### 2.8. Quantum Yield and Energy Conversion in Photosystem I

Regarding PSI (Figure 9a–d), plants subjected to drought stress exhibited lower curves for the quantum yield of photochemical energy conversion in PSI (Y(I)), the quantum yield of non-photochemical energy dissipation due to donor side limitation (Y(ND)), and the electron transport rate in PSI (ETR(I)). In contrast, the quantum yield of non-photochemical energy dissipation due to acceptor side limitation (Y(NA)) was significantly higher under drought conditions compared to control plants. The results indicated that the application of Si and SWE effectively modulated all parameters (Y(I), Y(ND), ETR(I), and Y(NA)), elevating them to levels that surpassed those of the control plants.

Under drought stress, P700ox, P700m, and P700m′ levels were significantly reduced in leaves exposed to elevated PAR intensities compared to well-watered controls (Figure 10a,b), reflecting a compromised PSI oxidation state. Silicon treatment not only alleviated this decline but fully restored and, in some cases, even exceeded control P700 levels, whereas SWE alone produced only a partial recovery, with P700 parameters remaining below those of unstressed plants. These findings underscore silicon’s pivotal role in preserving PSI integrity under severe water deficit and position seaweed extracts as useful, but less potent, modulators of photosynthetic resilience under drought.

### 2.9. Correlation Analysis

To identify possible correlations between treatments and the studied parameters, a correlation analysis was carried out (Table 1). The analysis of traits characterizing sesame plants subjected to drought stress, as well as to treatments with silicon (Si), seaweed extract (SWE), and their combined application, showed full agreement with the trait-by-trait analysis. These findings revealed the presence of both positive and negative correlations among the various evaluated parameters. In stressed plants, numerous negative correlations were observed for all growth-related parameters, including RL, SH, SD, RDW, SDW, WPDW, NL, and LA. Significant negative correlations were also found for photosynthetic pigment parameters, including chlorophyll *a* (chl. *a*), chlorophyll *b* (chl. *b*), and carotenoid content (car.). In addition, stressed plants exhibited negative correlations for gas exchange parameters such as net CO_2_ assimilation (*A*), stomatal conductance (*gs*), and transpiration rate (*E*), as well as for chlorophyll fluorescence and PSI parameters, such as the maximum quantum yield of PSII in dark-adapted samples (Fv/Fm), maximal fluorescence yield of dark-adapted samples with all PSI centers closed (P700m), and the oxidized state of PSI (P700ox). These correlations indicate that drought stress conditions tend to reduce all of these physiological and growth parameters. In contrast, treatments with Si, SWE, and their combination under drought stress resulted in several significant positive correlations for all the aforementioned parameters.

## 3. Discussion

The agricultural sector is actively seeking new strategies to increase crop yields in response to rising abiotic stressors exacerbated by global climate change. This study aimed to assess the effects of silicon, seaweed extracts, and the combined application of these two biostimulants on the activities of PSI and PSII, as well as gas exchange, in plants subjected to drought stress. While the primary focus was on the role of these biostimulants in enhancing photosynthesis under drought conditions, significant attention was also given to other key parameters, including growth indicators, pigment concentrations, and plant water status, such as relative water content and leaf water potential.

In the present study, the reduction in both fresh and dry biomass observed in stressed plants, compared to the control, may be attributed to decreased stomatal conductance. This reduction is a result of stomatal closure aimed at minimizing water loss through transpiration, which consequently limits CO_2_ assimilation during photosynthesis. This leads to a direct reduction in glucose biosynthesis [6,44]. Furthermore, drought stress in sesame was reflected in the decline of various growth traits, including leaf number, shoot length, and root length. Drought significantly impacts plant height, which is closely associated with cell enlargement and leaf senescence, due to inhibited cell growth, increased leaf loss, and impaired mitosis under drought conditions [45]. The decrease in root length is particularly notable, as roots are the first organs to detect water deprivation, rapidly sensing the stress and responding accordingly. Drought influences root length, diameter, surface area, tissue density, and biomass, likely due to the loss of turgor pressure. When turgor pressure is reduced, cells cannot expand properly, limiting their growth [46,47].

However, certain parts of the root system can still grow and access water from deeper soil layers, thus enhancing crop adaptation to drought stress [48]. The reduction in leaf number may function as a drought tolerance mechanism or a water conservation strategy [49]. In response to drought, plant leaves typically reduce their surface area, increase their thickness, and enhance tissue density. Previous studies have shown that changes in leaf area are mainly attributed to variations in leaf turgor pressure and canopy temperature, which reduce leaf turgor and photosynthetic rates, ultimately leading to a decrease in leaf area [50]. These findings are consistent with the observed reduction in sesame leaf area under drought stress conditions in the present study. This aligns with the results reported by Kouighat et al. (2024) [51] in sesame, as well as similar findings observed by Oukaltouma et al. (2022) [8] in faba bean (*Vicia faba* L.). Additionally, similar results were reported by Mouradi et al. (2016) [52] in alfalfa (*Medicago sativa* L.).

Leaves relative water content (LRWC) and leaf water potential (LWP) are critical parameters in plant water relations [53]. In *S. indicum* drought-stressed plants were significantly affected, corroborating the findings of El Boukhari et al. (2023) [54], who reported that drought treatment considerably reduced the relative water content in faba bean leaves. Similarly, in sweet basil (*Ocimum basilicum* L.), drought stress significantly decreased LRWC compared to the control group [55]. Farissi et al. (2013) [56] also observed that water deficiency exacerbated LRWC reduction in *Medicago sativa*, and Gong et al. (2012) [57] reported a decrease in leaf water potential in wheat plants under drought stress. This decrease is likely due to limited water absorption by roots caused by reduced soil water availability during drought conditions. Reduced water intake and increased water loss directly contribute to lower leaf RWC (*r* = −0.79) (Table 1) [28]. Furthermore, drought stress can cause cavitation and embolism in xylem vessels, which reduces the plant’s hydraulic conductivity. This, in turn, inhibits the transfer of water from roots to leaves, leading to a decline in leaf water potential (*r* = −0.98) (Table 1) [58,59].

In photosynthesis, chlorophyll is a central component of photosystems and plays a key role in regulating plant growth. Under drought stress, sesame plants exhibited a significant reduction in the concentrations of chl. *a* (*r* = −0.90), chl. *b* (*r* = −0.90), chl. *a* + chl. *b* (*r* = −0.90), and car. (*r* = −0.86) (Table 1). This decline is largely responsible for the reduction in plant growth and biomass production. Similar findings have been reported by Zonouri et al. (2014) [60] in grapes (*Vitis vinifera* L.), Batra et al. (2014) [61] in Mung bean (*Vigna radiata* L.), and Bijanzadeh et al. (2010) [62] in wheat. The decrease in chlorophyll levels under drought stress may be attributed to ultrastructural deformations in plastids, including the disruption of the thylakoid protein membranes. This leads to the disassembly of PSII, which is responsible for absorbing photons and facilitating electron transport [63,64].

This decrease could be attributed to stress-induced disruptions in pigment degradation or biosynthesis pathways [62]. Furthermore, Batra et al. (2014) [61] proposed that the overproduction of ROS under drought stress may also contribute to chlorophyll degradation. Alternatively, plants may close their stomata to reduce water loss through transpiration during water stress. However, this stomatal closure limits the absorption of CO_2_, which is essential for photosynthesis. As a result, this can contribute to reduced chlorophyll levels and a decrease in photosynthetic activity [65].

In addition to the composition of photosynthetic pigments, ABA plays a role in the regulation of stomatal closure under water stress, which correlates strongly with a reduction in stomatal conductance (*gs*). This consequently leads to lower net CO_2_ assimilation (*A*) and reduced transpiration (*E*). In this study, the results showed that plants exhibited reductions in *E* (*r* = −0.83), *gs* (*r* = −0.84), and *A* (*r* = −0.88) (Table 1). The effect of drought stress on gas exchange has also been observed in other studies [66,67,68].

Flexas et al. (2006) [69] demonstrated that when *gs* drops below a critical threshold, photosynthetic capacity is impaired, generally due to the simultaneous inhibition of photosynthetic enzymes and reductions in chlorophyll and protein contents. In line with this, CO_2_ assimilation is reduced under drought stress in chloroplasts, which limits the use of electrons for CO_2_ fixation in the Calvin cycle. This decreases excess electron production, which is crucial for photoprotection in plants [70]. When photosynthetic processes reach saturation, the over-reduction of electron transport components (ETR) leads to the transfer of electrons to oxygen at photosystem I, generating ROS that can cause oxidative damage to the photosynthetic apparatus [61].

Drought stress also affected the Fv/Fm ratio, which is a key parameter for detecting damage to photosystem II and potential photoinhibition [71]. The results of this study showed a significant reduction in the Fv/Fm ratio (*r* = −0.93) (Table 1), consistent with findings from other studies [72,73]. However, Oukarroum et al. (2009) [74] reported no significant changes in the Fv/Fm ratio under drought stress. Photoinhibition is characterized by a decrease in the Fv/Fm ratio, effective quantum yield of photosystem II photochemistry, and electron transport rate [75].

Under water stress, the photochemical efficiency of both photosystems is impaired due to disruptions in energy dissipation and electron transport. Y(II), which represents the fraction of energy utilized by PSII for photochemical conversion, decreases under drought conditions, indicating a reduced capacity to efficiently use light energy. The excess energy is primarily dissipated through non-photochemical mechanisms, including Y(NPQ), which corresponds to regulated thermal dissipation via photoprotective mechanisms, and Y(NO), which reflects passive and unregulated energy dissipation as fluorescence and heat. The increase in Y(NO) and the decrease in Y(NPQ) under water stress suggest a loss of PSII’s photoprotective capacity, leading to excess energy accumulation and an increased risk of photoinhibition [76,77]. Regarding PSI, water stress results in a decline in Y(I) and ETR(I), indicating reduced PSI photochemical efficiency. Additionally, a decrease in Y(ND) reflects donor-side limitations of PSI, likely due to restricted electron transport from PSII. In contrast, the increase in Y(NA) suggests excessive accumulation of electrons at the PSI acceptor side, caused by reduced Calvin cycle activity and impaired NADP+ regeneration, leading to an over-reduced state of PSI [74]. Furthermore, the reduction in P700ox and P700m suggests impaired PSI oxidation, exacerbating photo-oxidative damage.

In the search for strategies to mitigate the effects of drought stress on crops, the single or combined application of biostimulants has shown promise in improving growth, physiological, and photosynthetic traits. As a biostimulant, Patel et al. (2021) [78] found that silicon promotes plant development by regulating water and mineral nutrient levels, enhancing drought tolerance in peanut plants. It has also been demonstrated that Si supplementation may increase drought tolerance in plants through the accumulation of organic osmolytes, controlling osmotic potential [79]. Additionally, Si can promote the buildup of soluble carbohydrates and amino acids [80]. Regarding the impacts of aqueous seaweed-based bioproducts, they are gaining popularity in crops, enhancing the absorption of water by roots due to their unique bioactive components. These bioproducts have phytostimulatory properties that promote plant growth and yield. They also exhibit phytoelicitor action, inducing defensive responses in plants and contributing to resistance against abiotic stressors [81].

Consistent with these reported mechanisms, the combination of Si and SWE has a marked positive effect on the morphological attributes of *Sesamum indicum* L. under drought stress, with improvements observed in leaf number (*r* = 0.80), shoot length (*r* = 0.99), root length, and whole plant growth (*r* = 0.85) (Table 1). The alleviating effects of Si treatment on drought-induced growth impairments have been reported in other studies [82,83,84]. The mechanism underlying Si-induced improvement in drought tolerance is likely linked to enhanced water uptake within plant tissues and stimulation of cell elongation, resulting in greater fresh biomass [85]. Similarly, foliar application of SWE also mitigated the adverse effects of drought stress. The growth-promoting qualities of SWE can be attributed to its oligosaccharide, amino acid, and vitamin content [86]. Which contribute to improved growth indices and accelerated root and shoot development [87]. This is likely enhanced by the presence of growth hormones [88].

Building on the growth responses described above, biostimulant treatments, especially the combined Si + SWE, also improved foliar development and plant water status under drought. In the present study, Si biostimulant application under drought stress induced positive changes in the NL, LA, and LRWC. These results align with findings by Bounaouara et al. (2024) [33], who reported that Si incorporation into the growth medium of salt-stressed Sorghum (*Sorghum bicolor* L.) plants significantly increased leaf area and LRWC. Similar improvements were observed in *Medicago sativa* L. [34] and fenugreek (*Trigonella foenum-graecum* L.) [35]. This enhancement can be attributed to the stimulation of Si effects that improve root hydraulic conductivity and maintain cell wall integrity, thus contributing to improved water uptake in sesame plants under drought conditions. Additionally, Si mitigates stress by reducing transpirational water loss, which is the primary driving force for water movement from roots to leaves [89]. One proposed mechanism involves Si deposition in leaf epidermal cell walls, which reduces transpiration rate and improves tissue hydration [90,91]. Numerous studies have shown that Si’s effect on improving water status in plants is not solely related to modulation of transpiration rates but also involves the activity of aquaporins, which enhances root function and facilitates greater water and nutrient uptake [92,93,94].

In addition, studies by Lamsaadi et al. (2023) [32] on fenugreek and Laifa et al. (2023) [95] on sea barley (*Hordeum murinum* L.) suggested that Si supplementation increases total fresh weight, leaf fresh weight, and total chlorophyll, and enhances the rigidity of mature leaves. Regarding the effect of seaweed extract spray, and consistent with the Si + SWE predominant effect noted above, sesame plants treated with SWE showed a significantly larger leaf surface area (*r* = 0.99), a higher number of leaves (*r* = 0.90), and an increased leaf relative water content (*r* = 0.89) compared to stressed plants (Table 1). These findings are consistent with those of Nour et al. (2010) [96] on tomatoes (*Solanum lycopersicum* L.), where foliar application of seaweed extract resulted in a significant increase in leaf number and total leaf area. Furthermore, Battcharya et al. (2015) [97] reported that foliar spray with *Ascophyllum nodosum* (L.) Le Jol. enhanced the plants’ ability to retain water and grow by stimulating LRWC, leaf water potential, and leaf area expansion, thus improving water stress management and boosting photosynthesis. Although responses to seaweed products can vary among studies, in our experiment, the combined Si + SWE treatment markedly improved LRWC, leaf water potential, and leaf number in drought-stressed sesame.

In contrast, El Boukhari et al. (2023) [54] showed no difference in LRWC when seaweed extracts from different species, including the brown macroalga *Fucus spiralis* L., were applied to stressed plants. The improvement in leaf RWC observed here and elsewhere could be explained by the presence of osmoprotective substances in seaweed extracts, such as betaines, polyamines, and proline, which help sustain cell turgor and water content under drought conditions [21]. Seaweed extracts are also rich in phytohormones like cytokinins, auxins, and gibberellins, which stimulate cell division, expansion, and differentiation, thereby increasing leaf growth and photosynthesis capacity [98]. When both treatments were applied together, they resulted in significantly better outcomes for LRWC, leaf water potential, and leaf number in stressed sesame plants. Sujata et al. (2023) [88] found that the independent application of orthosilicic acid and SWE significantly improved LRWC, water potential, and osmotic potential.

Consistent with the above water status responses, according to Borrell et al. (2000) [99], increased chlorophyll content and SPAD values are associated with enhanced transpiration efficiency and productivity under limited water availability. Elevated chlorophyll levels in plants may help restore the leaves’ ability to photosynthesize, thus promoting growth [100]. In our study, a synergistic response between SWE and Si under drought stress in sesame was reflected in higher chlorophyll metrics. In this study, Si alone or in combination with SWE substantially restored the levels of chlorophyll a (chl. *a*) (*r* = 0.90), chlorophyll b (chl. *b*) (*r* = 0.90), total chlorophylls (chl. *a* + chl. *b*) (*r* = 0.90), carotenoids (car.) *(r* = 0.86), and SPAD values (*r* = 0.86) (Table 1). These results are in agreement with those reported by Saleh et al. (2024) [101] in potatoes (*Solanum tuberosum* L.). Additionally, Idoudi et al. (2024) [102] reported that Si treatment increased pigments, particularly chlorophyll a and total chlorophylls (chl. *a* + chl. *b*), under iron deficiency.

In this context, silicon plays a role in the photosynthetic process by enhancing it and preventing chlorophyll degradation. Silica bodies act as translucent “windows,” allowing light to pass through to the mesophyll, which in turn increases chlorophyll content [103]. In our sesame study, the application of seaweed extracts under drought stress in sesame plants increased the levels of pigments, including chlorophyll a (chl. *a*) (*r* = 0.96), chlorophyll b (chl. *b*) (*r* = 0.96), total chlorophylls (*r* = 0.96), carotenoids (*r* = 0.93), and SPAD values (*r* = 0.82) (Table 1). Chlorophyll levels in sesame plants increased when seaweed extract was applied after water stress, which may help reestablish the photosynthetic capacity of the leaves, contributing to growth recovery. Supporting this interpretation, recent studies have indicated that SWE treatment significantly improved photosynthetic traits. Kumari et al. (2011) [104] observed that seaweed extracts enhance chlorophyll and carotenoid contents in tomato plants by providing essential nutrients, promoting hormonal activity that stimulates growth, improving photosynthesis efficiency, enhancing antioxidant activity to cope with environmental stress, and increasing the activity of enzymes involved in pigment biosynthesis.

Gas exchange responses paralleled the pigment responses. Under stress conditions, the supply of Si significantly enhanced the transpiration rate, likely driven by increased stomatal conductance (*gs*) activity, which helps maintain a steady state of photosynthetic CO_2_ assimilation. In this study, Si (and Si + SWE) significantly increased transpiration (*E*), *gs*, and net CO_2_ assimilation (*A*). Similar results were reported by Verma et al. (2021) [22] in sugarcane and Maghsoudi et al. (2016) [83] in wheat (*Triticum aestivum* L.), supporting the role of Si in mitigating drought-induced declines in photosynthetic efficiency. Gas exchange responses followed the same pattern seen for growth: treatments containing seaweed extract (SWE and Si + SWE) improved stomatal conductance, transpiration, and CO_2_ assimilation in drought-stressed sesame.

Moreover, the application of seaweed extracts alone also enhanced gas exchange under stress conditions, as evidenced by increased *gs* (*r* = 0.98), *E* (*r* = 0.98) and *A* (*r* = 0.91) (Table 1). Consistent with this, refs. [41,105] found that under abiotic stress, the application of SWE improves leaf gas exchange, primarily by reducing stomatal closure. Additionally, the application of *Ascophyllum nodosum* (L.) Le Jol. SWE under drought stress enhances gas exchange by increasing stomatal conductance and reducing stomatal limitation, which leads to improved CO_2_ assimilation [106].

To further evaluate how these treatments influenced photosynthetic machinery, we assessed PSI and PSII performance under drought. Overall, treatments that included SWE whether applied alone or in combination with Si produced more favorable photochemical responses than Si alone. For SWE application under stress conditions, the photosynthetic status was evaluated by assessing the activities of PSI and PSII. Overall, the effect of seaweed extracts, either alone or in combination with Si, produced more favorable results than Si alone. SWE application induced a significant increase in Fv/Fm (*r* = 0.90), P700ox (*r* = 0.64), and P700m (*r* = 0.95) (Table 1). A similar trend was observed by Santaniello et al. (2017) [107], who found that the application of *Ascophyllum nodosum* (L.) Le Jol. extracts under drought stress enhances both the maximum quantum yield of photosystem II (Fv/Fm) and the redox state of P700ox and P700m, indicating improved photosynthetic efficiency and electron transfer. These improvements are attributed to protective mechanisms that stabilize photosynthetic complexes and optimize electron transport, allowing plants to maintain photosynthesis under drought stress.

Our results demonstrated that seaweed extracts containing treatments (SWE and Si + SWE) had the most significant effect on the restoration of Y(II) values, the increase in Y(NPQ), the enhancement of ETR(II), and the reduction in Y(NO) as light intensities increased under water stress. According to our findings, Santaniello et al. (2017) [107] reported that seaweed extracts (SWE) enhance the efficiency of photosystem II (PSII) by increasing Y(II) and the electron transport rate (ETR), thereby optimizing the conversion of light energy into chemical energy. They also stimulate Y(NPQ) by activating photoprotective mechanisms, facilitating the dissipation of excess excitation energy as heat and helping preserve PSII integrity. Furthermore, the reduction in Y(NO) indicates a decrease in oxidative stress due to enhanced antioxidant defenses, which limit the accumulation of reactive oxygen species (ROS) and protect PSII from photoinhibition. Similarly, seaweed extracts improve PSII efficiency by providing bioactive compounds that enhance stress tolerance, water retention, and nutrient availability, leading to improved light reactions under drought [108].

The combination of seaweed extracts and Si produced beneficial though sometimes intermediate results, while Si alone showed positive effects, albeit to a lesser degree. The increase in Y(NPQ) likely reflects a protective mechanism to mitigate ROS production and maintain the balance between light absorption and utilization [30]. In this context, the elevated Y(NPQ) and reduced Y(NO) resulted in enhanced non-photochemical quenching (NPQ), which likely contributed to the protection of PSII from photoinhibition by limiting excessive electron flow to PSII via linear electron transport [33].

Regarding PSI, our data indicated that seaweed extracts under drought stress exhibited superior results in PSI compared to Si alone. This was evidenced by similar trends in Y(I), Y(ND), and ETR(I), as well as reduced Y(NA) energy dissipation, which reflected the extent of donor-side and acceptor-side restrictions to PSI oxidation, respectively. These responses suggest that Si may also contribute by enhancing PSI oxidation and protecting PSI from ROS-induced photoinhibition [109].

## 4. Materials and Methods

### 4.1. Plant Material

A Moroccan type of sesame (*Sesamum indicum* L.) was the plant material employed in this investigation. This type is one of the most often employed by local farmers due to its great degree of adaptation to local conditions. The National Institute of Agronomic Research (INRA-Morocco) provided the sesame seeds.

### 4.2. Plant Collection Sites and Physicochemical and Biochemical Properties of the Brown Seaweed Algae

*Sargassum muticum* (Yendo) Fensholt was collected from the Atlantic coast of Morocco, near Mohammedia (33°41′9.85″ N, 7°22′58.73″ W). In the laboratory, seaweed samples were thoroughly washed with distilled water several times to ensure the removal of impurities. After the cleaning process, the thalli were left to air dry for five days until completely dried. After being dried, the seaweed material was chopped into tiny pieces and milled into a fine powder. For the preparation of seaweed extracts (SWE), see Ajjabi et al. (2019) [110]. After preparation, the aqueous algae extract was characterized, and its physicochemical and biochemical composition is summarized in Table 2.

### 4.3. Experimental Procedure for Drought Stress

At the beginning of the experiment, for one month, the pots were watered every three days with 150 mL of water until the plants reached the end of their initial vegetative growth phase. Following this, two irrigation regimes were applied, representing opposing treatments: full irrigation (control) throughout the plant’s life cycle and drought stress (DS). In order to apply the drought stress treatment, plants were irrigated every six days, whereas control plants were watered every three days, using the same volume of water per irrigation. This irrigation regime corresponds to a 50% reduction in water supply for the stressed plants compared to the non-stressed ones. The treatment was maintained consistently over a sixty-three-day period until harvest.

### 4.4. Experimental Design

The experiment was conducted at the Centre of Biotechnology of Borj-Cedria (Northeastern Tunisia, 36°42′32.9″ N, 10°25′40.9″ E) in a growth chamber under controlled conditions, with daytime temperatures maintained at 25 ± 2 °C, nighttime temperatures at 16 ± 2 °C, relative humidity between 60% and 80%, and a 16 h photoperiod (250 µmol photons s^−1^ m^−2^). Seeds were sown in plastic pots measuring 10 cm in height and 12 cm in diameter, containing 500 g of a sand–peat–soil mixture (1:1:1 *v*/*v*/*v*). Two weeks after sowing, three homogeneous seedlings were maintained and irrigated weekly with Hewitt nutrient solution [111]. Each solution delivered the full set of macro- and micronutrients required for healthy plant growth. The following macronutrient concentrations (in mM) were used: Ca(NO_3_)_2_, 3.0; KNO_3_, 2.0; NH_4_NO_3_, 1.6; KH_2_PO_4_, 0.6; K_2_HPO_4_, 1.5; and MgSO_4_·7H_2_O. The following micronutrient concentrations (in μM) were added: 9.1 MnCl_2_·4H_2_O, 1.1 CuSO_4_·5H_2_O, 0.8 ZnSO_4_·7H_2_O, 0.05 H_3_BO_3_, and 0.02 (NH_4_)_6_Mo_7_O_24_·4H_2_O. The pH of the nutrient solution was adjusted to 6.4. The nutrient solution was first diluted to 1/4 of its standard concentration, then to 1/2 of the standard concentration, followed by treatment C. After 24 days, the first stress application was followed by dividing the plants into two groups, each receiving one of the two irrigation regimes: full irrigation (C) and drought stress (DS). After a 9-day stress application (at 47 days of age), plants were separated into four groups for each irrigation regime: Group one (C, control) stayed on the standard nutrient solution, whereas group two was watered with the same solution and foliar spray with 1 mM of sodium silicate (Si: Na_2_SiO_3_). The third group received foliar spray in addition to a nutritional solution with 10% of seaweed extracts (SWE), and the fourth group received a nutrient solution supplemented with foliar spray with 1 mM Na_2_SiO_3_ and 10% SWE (Na_2_SiO_3_ + SWE). The plants were treated five times, twice a week, with each pot receiving 20 mL of foliar spray. Each pot held three plants, and each treatment was repeated ten times, for a total of 80 pots and 240 plants. To ensure uniform growth, the culture conditions (light, humidity, and temperature) as well as the orientation and the arrangement of the pots were systematically changed biennially.

### 4.5. Morphological Indicators of Growth

Plants were removed and divided into roots and shoots following a 64-day treatment period. To remove extracellular nutrients, root samples were gently rinsed in an ice-cold 0.01 M HCl bath. They were then rinsed with distilled water and blotted on filter paper. Using a centimeter-graduated ruler, stem diameter (SD), shoot length (SL), and root length (RL) were measured along with the number of leaves to quantify drought stress responses. To determine the dry weight (DW), root DW (RDW), shoot DW (SDW), and whole plant DW (WPDW), samples were oven dried to constant mass at 60 °C. Three leaves from each of the three plants in each treatment were used to calculate the leaf area in Mesurim (version 3.4.4.0) [56].

### 4.6. Leaf Relative Water Content (LRWC)

Following the procedure of Barrs and Weatherley (1962) [112], we determined leaf water status. At the third-leaf stage, one mature leaf from each of three plants per treatment was sampled. Disks punched from these leaves were weighed immediately to record fresh weight (FW), floated on distilled water at room temperature for 24 h to reach full turgor (TW), and then oven-dried at 70 °C for 24 h to obtain dry weight (DW).RWC = [(FW − DW)/(TW − DW)] × 100,(1)

### 4.7. Leaf Water Potential (LWP) Determination

A pressure chamber (Model 600, PMS Instrument Co., Albany, OR, USA) was used to measure leaf water potential (LWP). To determine LWP, we promptly chose fully expanded, symptom-free leaves from the same stem for each treatment. Measurements were taken in triplicate and the mean values are reported.

### 4.8. Photosynthetic Pigment Content

We adopted Lichtenthaler’s (1987) method to extract photosynthetic pigments [113]. Fifty milligrams of fresh leaf disks was placed in vials containing 3 mL of 80% acetone and stored for 72 h at 4 °C in the dark. The absorbance of the extract was measured using a UV-visible spectrophotometer (Specord 210 Plus, Analytik Jena, Jena, Germany) at 470, 646, and 663 nm.Chlorophyll *a* (μg mL^−1^) = (12.21 × A663) − (2.81 × A646); chlorophyll *b* (μg mL^−1^) = (20.13 × A646) − (5.03 × A663); and carotenoids (μg mL^−1^) = (1000 × A470 − 3.27 × [chl. *a*] − 104 × [chl. *b*])/229
where A470, A646, and A663 represent absorbance values at the corresponding wavelengths.

### 4.9. Photosynthetic Gas Exchange

A portable LC Pro gas analyzer (LC pro+, ADC BioScientific Ltd., Hoddesdon, UK) was used to measure photosynthetic gas exchange on fully expanded leaves during specific hours of the day, from 10:00 to 12:00, while taking into account a number of environmental factors. At a photosynthetically active radiation (PAR) of 1000 µmol m^−2^ s^−1^, with ambient CO_2_ concentrations around 450 μmol mol^−1^, leaf chamber temperatures around 25 °C, and atmospheric pressures of approximately 1022 mBar, measurements were conducted in saturating light. Net CO_2_ assimilation (*A*), stomatal conductance (*gs*), and transpiration rate (*E*) were among the data gathered.

### 4.10. Assessment of Photosystem Activities

A Dual-PAM-100 fluorometer (Heinz Walz, Efeltrich, Germany) was used to measure photosystem activity according to Klughammer and Schreiber (2008a, b) [114,115]. To calculate the maximum quantum yield of photosystem II (PSII) as Fv/Fm, *Sesamum indicum* L. was dark-adapted for 30 min to determine minimal fluorescence (F0) and maximal fluorescence (Fm), enabling the calculation of Fv/Fm. The measurement process began with determining F0 under dark conditions, followed by a saturating light flash to record Fm, at which point all PSII reaction centers were closed. Subsequently, the leaves were exposed to various intensities of actinic light (0 to 1017 μmol photons m^−2^ s^−1^) to start electron transport and quantify important parameters, such as the yield of regulated non-photochemical energy dissipation (Y(NPQ)), non-regulated dissipation (Y(NO)), and photochemical energy conversion in PSII (Y(II)). To evaluate photochemical energy conversion (Y(I)) and non-photochemical energy dissipation on both the donor and acceptor sides (Y(ND) and Y(NA)), P700 absorbance was measured for photosystem I (PSI) at dual wavelengths (830 and 875 nm). This comprehensive method allowed the simultaneous assessment of the P700 redox status and chlorophyll fluorescence, providing valuable information about the physiological condition of the plants and the effectiveness of the photochemical and biochemical processes involved in photosynthesis.

### 4.11. Statistical Analysis

All statistical analyses in this study were performed using XLSTAT software version 2014 (www.xlstat.com), developed by Addinsoft, France. Duncan’s Multiple Range Test was used to compare treatment means at a 95% confidence level, while Pearson correlation analysis was conducted to assess the relationships between variables.

## 5. Conclusions

This study underscores the susceptibility of sesame to drought stress, which detrimentally impacts its growth, water relations, and physiological processes. Encouragingly, the findings demonstrate that applying biostimulants such as Si and *Sargassum muticum* (Yendo) Fensholt SWE can markedly enhance the plant’s tolerance to water-limited conditions. When applied separately, each biostimulant exhibited distinct advantages. SWE was notably effective in improving photosynthetic pigment content, gas exchange, and photosystem II efficiency while also stimulating root branching, whereas Si played a more prominent role in stabilizing photosystem I and promoting root elongation. Under well-watered conditions, both treatments helped maintain vigorous growth, sustained photosynthetic activity, and optimal water status, highlighting their potential to enhance plant vigor beyond merely alleviating stress. Notably, both in the presence and absence of drought stress, the combined application of Si and SWE led to the most significant physiological and morphological improvements, suggesting a synergistic and complementary interaction between the two treatments. This study thus outlines two practical approaches for biostimulant use in sesame cultivation: employing Si or SWE individually to enhance targeted physiological functions according to agronomic goals or applying them together to achieve broader enhancements in growth, photosynthesis, stomatal conductance, and resilience to stress. Beyond their physiological benefits, these environmentally friendly biostimulants could contribute to more sustainable agricultural practices, especially in arid and semi-arid regions where water scarcity is intensifying. Incorporating these biostimulants into crop management could therefore be an effective strategy for improving plant resilience under both optimal and climate-induced stress conditions.

## Figures and Tables

**Figure 1 plants-14-02358-f001:**
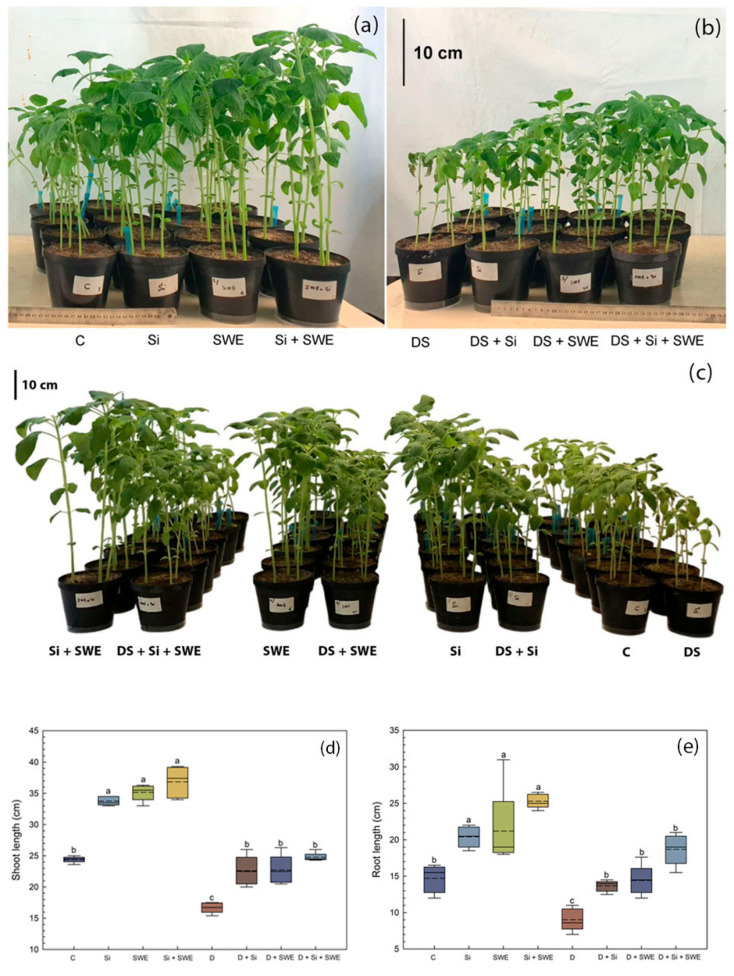
Morphological responses of sesame plants (n = 10 per treatment) to biostimulant treatments under the following: (**a**) well-watered conditions (control) with silicon (Si, 1 mM Na_2_SiO_3_), seaweed extract (SWE, 10% *Sargassum muticum* (Yendo) Fensholt), or their combination (Si + SWE); (**b**) drought stress conditions (DS) with equivalent treatments; and (**c**) comparative view demonstrating treatment mediated drought mitigation. (**d**) Box plot showing shoot length and (**e**) root length of sesame plants treated with C (control), Si, SWE, Si + SWE, DS (drought stress), DS + Si, DS + SWE, and DS + Si + SWE. The middle line within the box represents the median, while the dashed line indicates the mean. Data are presented as mean ± SE (n = 5). Significant differences between treatments were determined using Duncan’s test (*p* < 0.05) for each parameter and indicated by different lowercase letters above the box.

**Figure 2 plants-14-02358-f002:**
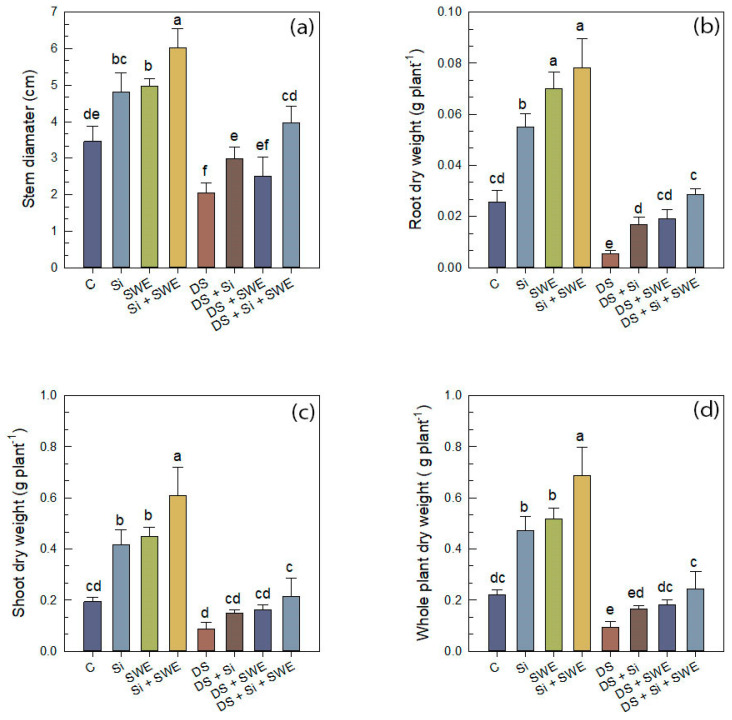
Effect of applied drought stress on growth indicators of control plants and plants treated with silicon (1 mM Si), seaweed extract (10% SWE), and their combination (1 mM Si + 10% SWE): (**a**) stem diameter; (**b**) root dry weight; (**c**) shoot dry weight; (**d**) whole plant dry weight. Data are presented as mean ± SE (n = 5). Significant differences between treatments were determined using Duncan’s test (*p* < 0.05) for each parameter and indicated by different lowercase letters above the bars.

**Figure 3 plants-14-02358-f003:**
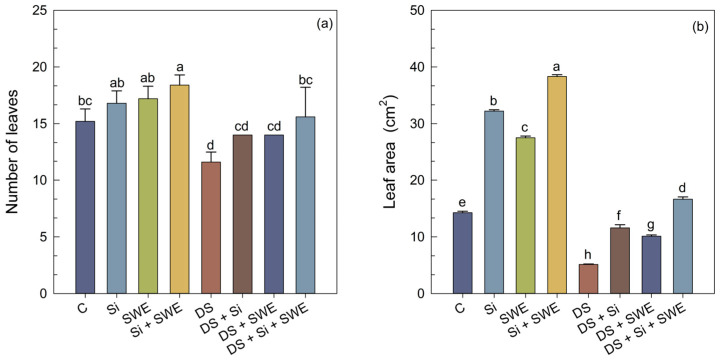
Effect of applied drought stress on morphological indicators of control sesame plants and those treated with silicon (1 mM Si), seaweed extract (10% SWE), and their combination (1 mM Si + 10% SWE): (**a**) number of leaves; (**b**) leaf area. Data are presented as mean ± SE (n = 5). Significant differences between treatments were determined using Duncan’s test (*p* < 0.05) for each and indicated by different lowercase letters above the bars.

**Figure 4 plants-14-02358-f004:**
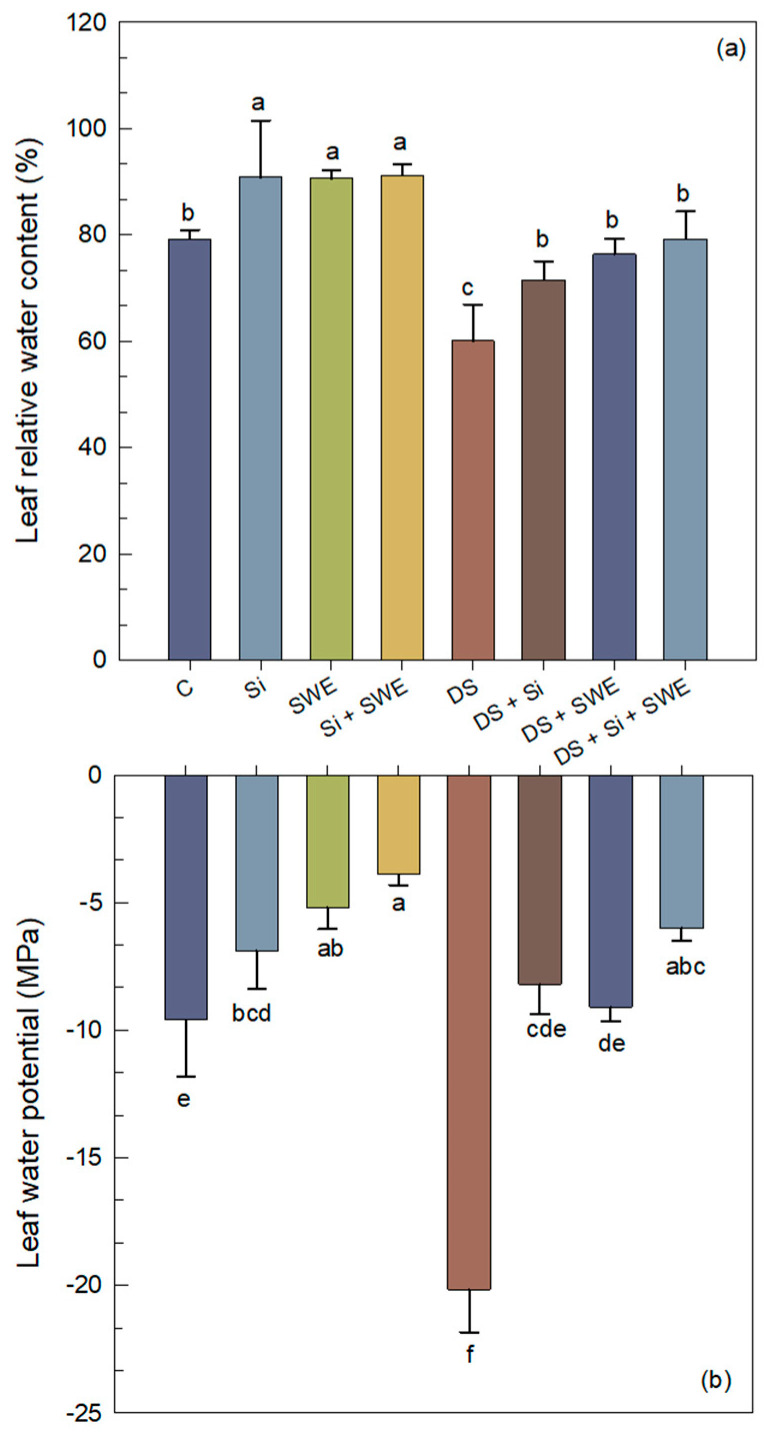
Effect of applied drought stress on physiological parameters of control sesame plants and those treated with silicon (1 mM Si), seaweed extract (10% SWE), and their combination (1 mM Si + 10% SWE): (**a**) leaf relative water content; (**b**) leaf water potential. Data are presented as mean ± SE (n = 5). Significant differences between treatments were determined using Duncan’s test (*p* < 0.05) for each parameter and indicated by different lowercase letters above the bars.

**Figure 5 plants-14-02358-f005:**
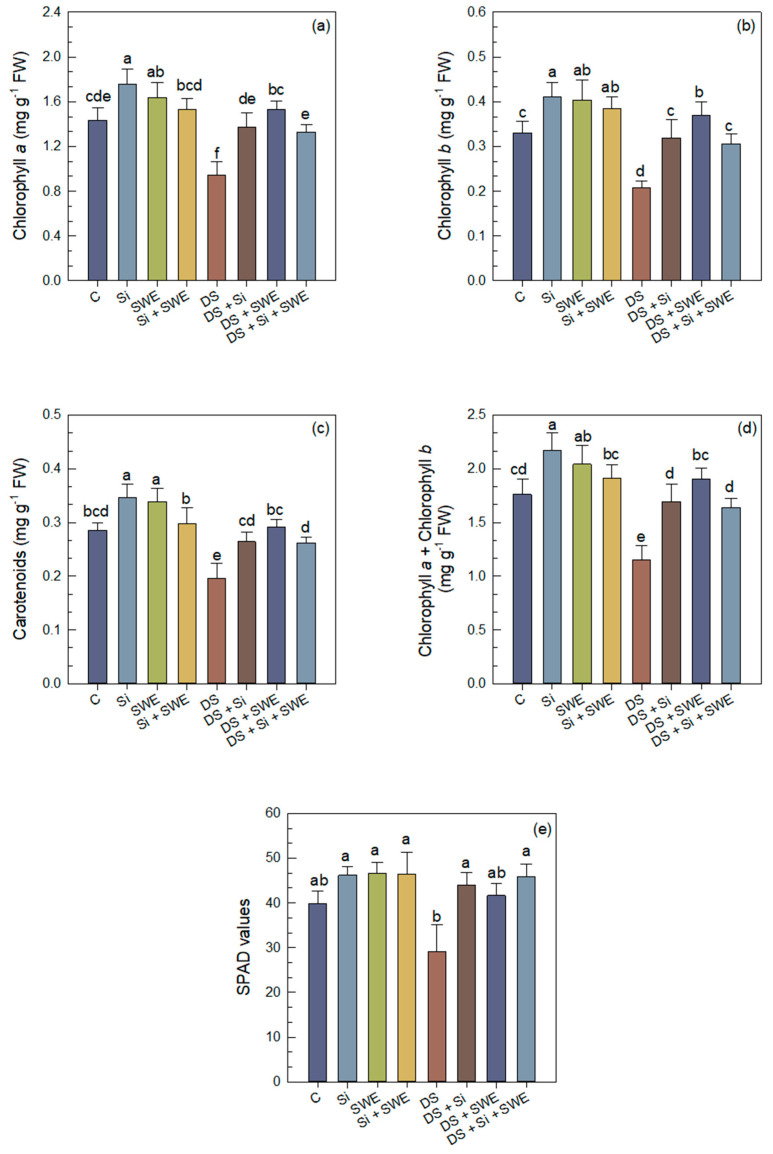
Effect of applied drought stress on photosynthetic pigment content in control sesame plants and those treated with silicon (1 mM Si), seaweed extract (10% SWE), and their combination (1 mM Si + 10% SWE): (**a**) chl. *a*; (**b**) chl. *b*; (**c**) car.; (**d**) total chl. (*a + b*); (**e**) SPAD values. Data are presented as mean ± SE (n = 5). Significant differences between treatments were determined using Duncan’s test (*p* < 0.05) for each parameter and indicated by different lowercase letters above the bars.

**Figure 6 plants-14-02358-f006:**
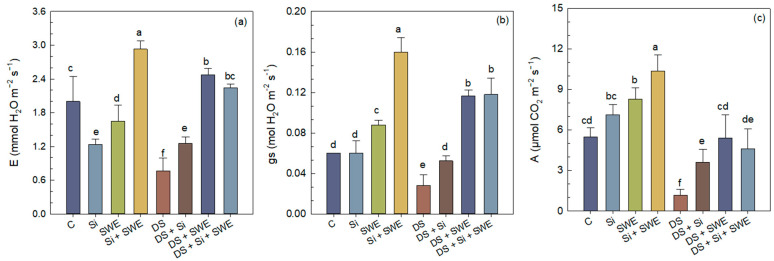
Effect of applied drought stress on photosynthetic gas exchange parameters in control sesame plants and those treated with silicon (1 mM Si), seaweed extract (10% SWE), and their combination (1 mM Si + 10% SWE): (**a**) *E*: transpiration rate; (**b**) *gs*: stomatal conductance; (**c**) *A*: net CO_2_ assimilation. Data are presented as mean ± SE (n = 5). Significant differences between treatments were determined using Duncan’s test (*p* < 0.05) for each parameter and indicated by different lowercase letters above the bars.

**Figure 7 plants-14-02358-f007:**
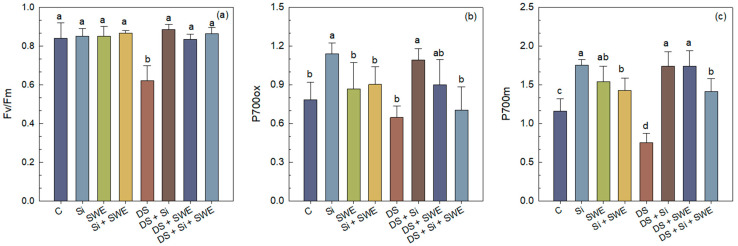
Effect of applied drought stress on fluorescence parameters of PSII in dark-adapted leaves (dark test) of control sesame plants and those treated with silicon (1 mM Si), seaweed extract (10% SWE), and their combination (1 mM Si + 10% SWE): (**a**) Fv/Fm: maximal photosystem II (PSII) quantum yield of dark-adapted samples; (**b**) P700ox: oxidized photosystem I (PSI); (**c**) P700m: maximal fluorescence yield of dark-adapted samples with all PSI centers closed. Data are presented as mean ± SE (n = 5). Significant differences between treatments were determined using Duncan’s test (*p* < 0.05) for each parameter and indicated by different lowercase letters above the bars.

**Figure 8 plants-14-02358-f008:**
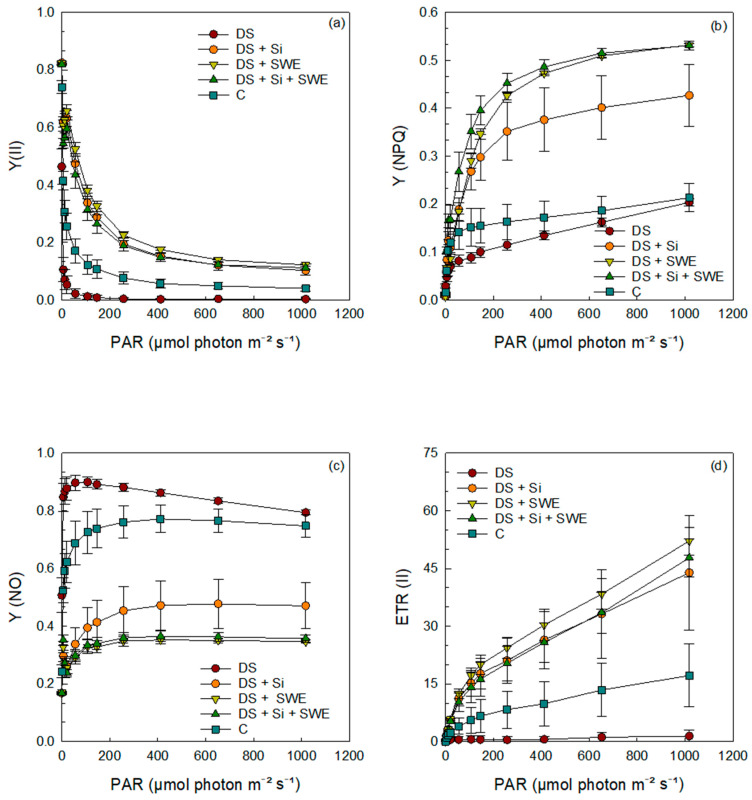
Effect of applied drought stress on PSII yield activity in light-adapted leaves of control sesame plants and those treated with silicon (1 mM Si), seaweed extract (10% SWE), and their combination (1 mM Si + 10% SWE) after a minimum exposure of 30 s at each light intensity (PAR: photosynthetically active radiation): (**a**) Y(II): quantum yield of photochemical energy conversion in PSII; (**b**) Y(NPQ): quantum yield of regulated non-photochemical energy dissipation in PSII; (**c**) Y(NO): quantum yield of non-regulated non-photochemical energy dissipation in PSII; (**d**) ETR(II): electron transfer rate in PSII. Data are presented as mean ± SE (n = 5).

**Figure 9 plants-14-02358-f009:**
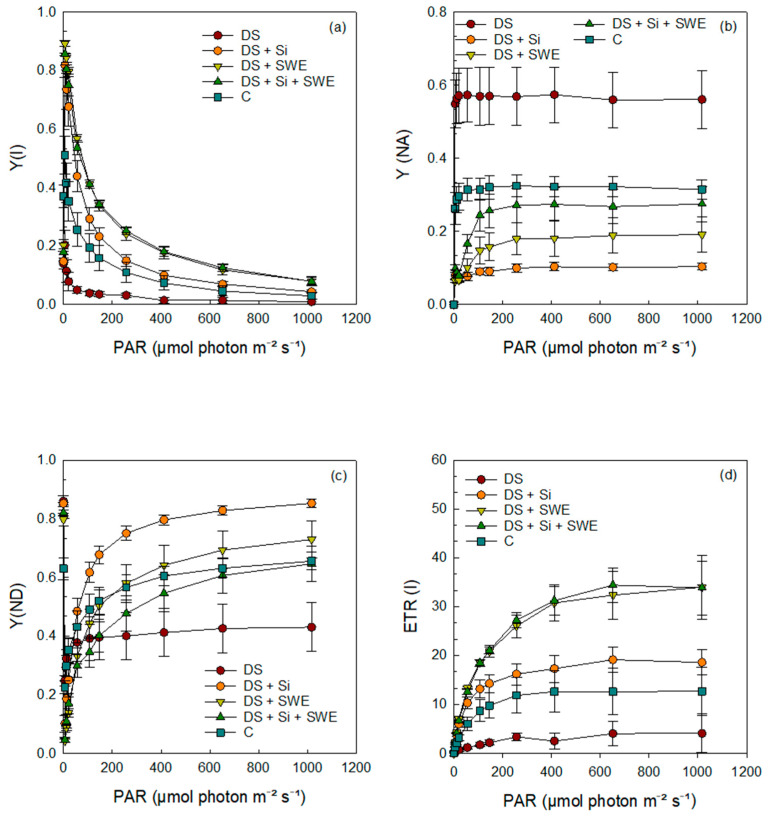
Effect of applied drought stress on PSI yield activity in light-adapted leaves of control sesame plants and those treated with silicon (1 mM Si), seaweed extract (10% SWE), and their combination (1 mM Si + 10% SWE) after a minimum exposure of 30 s at each light intensity (PAR: photosynthetically active radiation): (**a**) Y(I): quantum yield of photochemical energy conversion in PSI; (**b**) Y(NA): quantum yield of non-photochemical energy dissipation limited by the acceptor side in PSI; (**c**) Y(ND): quantum yield of non-photochemical energy dissipation limited by the donor side in PSI; (**d**) ETR(I): electron transfer rate in PSI. Data are presented as mean ± SE (n = 5).

**Figure 10 plants-14-02358-f010:**
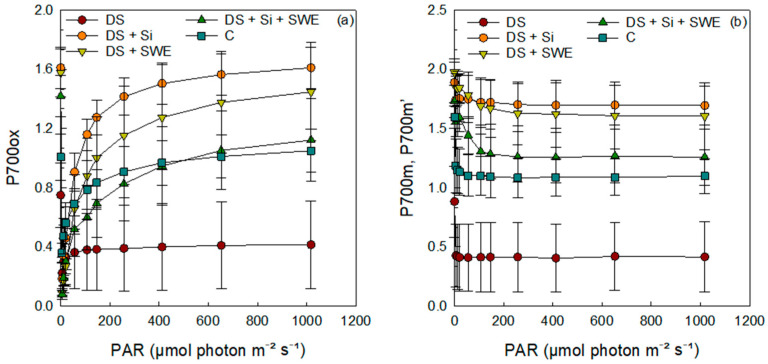
Effect of drought stress on PSI fluorescence parameters in light-adapted leaves (light test) of control plants and those treated with silicon (Si, 1 mM), seaweed extract (SWE, 10%), or their combination (Si + SWE). Measurements were performed after a minimum exposure of 30 s at each light intensity (PAR, photosynthetically active radiation): (**a**) P700ox, oxidation state of PSI; (**b**) P700m, maximal fluorescence yield of the dark-adapted sample with all PSI centers closed; P700m, maximal fluorescence yield of the illuminated sample with all PSI centers closed. Data are presented as mean ± SE (n = 5).

**Table 1 plants-14-02358-t001:** Pearson’s correlation matrix analyzing drought stress (DS) treatments (in the absence and presence of silicon, seaweed extracts, and their combination) and the different parameters studied. The values in the table represent the correlation coefficients (*r*) between the different variables. *r* values marked with * indicate statistically significant correlations (*alpha* < 0.05). These correlation coefficients were centered around their means and normalized with a standard deviation of 1. Negative correlations are shown in blue, while positive correlations are in red. RL: root length; SH: shoot length; SD: stem diameter; RDW: root dry weight; SDW: shoot dry weight; WPDW: whole plant dry weight; NL: number of leaves; LA: leaf area; chl. *a*: chlorophyll a content; chl. *b*: chlorophyll b content; Car.: carotenoid content; *A*: net CO_2_ assimilation; *gs*: stomatal conductance; *E*: transpiration rate; Fv/Fm (PSII): quantum yield of dark-adapted samples; P700m: maximal fluorescence yield of dark-adapted samples with all PSI centers closed; P700ox: oxidized PSI.

	Variables	DS	DS + Si	DS	DS + SWE	DS	DS + Si + SWE
−1	RL	−0.91 *	0.91 *	−0.86 *	0.86 *	−0.95 *	0.95 *
−0.9	SL	−0.88 *	0.88 *	−0.89 *	0.89 *	−0.99 *	0.99 *
−0.8	SD	−0.86 *	0.86 *	−0.51	0.51	−0.94 *	0.94 *
−0.7	RDW	−0.94 *	0.94 *	−0.93 *	0.93 *	−0.99 *	0.99 *
−0.6	SDW	−0.88 *	0.88 *	−0.89 *	0.89 *	−0.80 *	0.80 *
−0.5	WPDW	−0.91 *	0.91 *	−0.91 *	0.91 *	−0.85 *	0.85 *
−0.4	NL	−0.90 *	0.90 *	−0.90 *	0.90 *	−0.75 *	0.75 *
−0.3	LA	−0.99 *	0.99 *	−0.99 *	0.99 *	−0.99 *	0.99 *
−0.2	LRWC	−0.79 *	0.79 *	−0.89 *	0.89 *	−0.88 *	0.88 *
−0.1	LWP	−0.98 *	0.98 *	−0.98 *	0.98 *	−0.99 *	0.99 *
0	Chl. *a*	−0.90 *	0.90 *	−0.96 *	0.96 *	−0.92 *	0.92 *
0.1	Chl. *b*	−0.90 *	0.90 *	−0.96 *	0.96 *	−0.93 *	0.93 *
0.2	Car.	−0.86 *	0.86 *	−0.93 *	0.93 *	−0.89 *	0.89 *
0.3	Chl. *a* + Chl. *b*	−0.90 *	0.90 *	−0.96 *	0.96 *	−0.93 *	0.93 *
0.4	SPAD	−0.86 *	0.86 *	−0.82 *	0.82 *	−0.89 *	0.89 *
0.5	*E*	−0.83 *	0.83 *	−0.98 *	0.98 *	−0.98 *	0.98 *
0.6	*gs*	−0.84 *	0.84 *	−0.98 *	0.98 *	−0.96 *	0.96 *
0.7	*A*	−0.88 *	0.88 *	−0.91 *	0.91 *	−0.86 *	0.86 *
0.8	Fv/Fm	−0.93 *	0.93 *	−0.90 *	0.90 *	−0.92 *	0.92 *
0.9	P 700ox	−0.94 *	0.94 *	−0.64	0.64	−0.20	0.20
1	P700 m	−0.96 *	0.96 *	−0.95 *	0.95 *	−0.92 *	0.92 *

**Table 2 plants-14-02358-t002:** Physicochemical and biochemical properties of *Sargassum muticum* (Yendo) Fensholt extracts used in this study.

Characteristics	Values
pH	7.3 ± 0.10
Electrical conductivity (µS/cm)	1.196 ± 2.03
Total soluble sugars (mg/g DW)	31.17 ± 0.03
Potassium (mg/L)	30.43 ± 0.03
Magnesium (m/L)	27.22 ± 0.002
Calcium (mg/L)	43.8 ± 0.004
Chloride (mg/L)	40.54 ± 0.001
Nitrate (mg/L)	5.83 ± 0.02
Sulfate (mg/L)	22.47 ± 0.04
Sodium (mg/L)	149.11 ± 0.00
Total polyphenol content (mg/g DW)	24.31 ± 0.00
Flavonoid content (mg/g DW)	3.61 ± 1.14
Phosphorus content (mg/g DW)	34.34 ± 0.10
Proline content (mg/g DW)	2.90 ± 0.52
Total antioxidant activity (mg GAE/g DW)	28.79 ± 3.31
Protein content (mg/g DW)	18.73 ± 0.10

## Data Availability

The original contributions presented in this study are included in the article. Further inquiries can be directed to the corresponding author.

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
