# Peer review of "Biostimulatory Effects of Foliar Application of Silicon and Sargassum muticum Extracts on Sesame Under Drought Stress Conditions"

_plants, 2025, doi:10.3390/plants14152358_

Round 1

Reviewer 1 Report

Comments and Suggestions for Authors

The authors present an interesting work on the impact of different biostimulants, alone or in combination, on the response of sesame plants to drought stress.

However, the authors should carefully review the manuscript to address major shortcomings, particularly in the results and discussion.  

Major revision

The authors should carefully review the manuscript, especially in the results section. This is because when reading this work, more emphasis is placed on the effect of drought on the plant than on the effect of the biostimulants applied, alone or in combination, in response to drought. Throughout the results, the results obtained from drought are presented first and with greater emphasis, and the data obtained after the application of biostimulants are not compared with the control treatment.

Furthermore, in the discussion, more emphasis is placed on the effects of drought on sesame plants than on the effect of the biostimulants, which are truly the object of study. It is particularly striking that in the first part of the discussion (lines 398-505), only the drought results are discussed. It would be advisable to restructure the discussion to give a prominent role to the biostimulants and, especially, to the combined SI+SWE treatment. Furthermore, it would be necessary to compare the results presented with other results related to biostimulants of this type or with biostimulant formulations containing different extracts/compounds of different origins.

Lines 364-396. The authors should also pay special attention to the correlation results of the DS treatment. Based on what has been explained throughout the work, it is assumed that the DS results are always the same, and that three different trials were not performed. However, it is not clear why the DS results of the correlation are different in the different columns. If the correlation was performed with DS vs. DS+Si, DS vs. DS+SWE, and DS+Si+SWE, this should be clearly indicated in the text.

It is recommended, in addition to all the results presented in the manuscript, to also include the results of the impact of the tested biostimulants on sesame production, if available. These data are expected throughout the manuscript since the sesame production cycle is short and it is easy to quantify the g of seeds obtained.

Minor revision

Lines 5-6: Add surnames in the same format, preferably in lowercase except for the first letter.

Line 55: Include sugar between "such as" and "proline."

Line 66: Introduce the common name.

Line 74: Replace "which makes" with "making."

Line 94: Delete "reactive oxygen species," since ROS has already been defined in lines 51-52.

Line 97: Introduce "reactive water content" (RWC) [8].

Lines 117-124: Better define the objectives of the work.

Line 129: Delete "silicon" and "seaweed," since the acronyms have already been defined in line 83 for silicon, and "SWE" in line 101.

Line 135. Complement Figure 1 with a boxplot that shows the total plant size, including root and shoot length.

Line 136. Correct the figure caption where Figure 2 is indicated, but is Figure 1.

Line 138. Italicize the specific name Sargassum muticum.

Lines 139-140. Indicate the acronyms for control and drought stress, as indicated in the text, also in the figure caption.

Line 154. Diameter, its acronym, has already been defined on line 145.

Lines 160-162. Indicate which figures the following text refers to: Drought stress also caused a significant (p < 0.05) reduction in SDW (Figure 2e), while the effect on RDW (Figure 2d) was statistically insignificant, with reductions of 55% and 81%, respectively. The whole plant dry weight (WPDW) (Figure 2f)…

Line 165. Correct p < 0.05 to p > 0.05

Lines 176-177. Introduce Figure 3b after number of leaves and delete a-b in Figure 3a-b since it is assumed that these are the two images.

Line 189. Capitalize the L in Leaf and lowercase the initials of water and content.

Beginning of line 234, add under control conditions after alone.

Lines 279-284. Improve the results presented in this paragraph. Add the results for both the control and stress conditions, as well as those obtained from the individual and combined treatments.

Line 288. Add under drought stress after to control levels.

Lines 310, 316, and 330. Define the meanings of PSII(II), Y(NPQ), ETR (II), Y(NO), Y(I), Y(ND), ETR(I), and Y(NA).

Line 313. The acronyms PAR have been defined differently in lines 313 and 346-347.

Line 328. Delete the space between 2. and 8 at the beginning of the title.

Line 363. Indicate that the correlation analysis section is section 2.9 and delete the colon at the end of the title.

Line 365. Correct Table 2 as Table 1.

Lines 371 and 372. The acronyms defined in these lines have already been previously defined, and there is no need to redefine each acronym.

Line 398. Delete the "Authors should" from the beginning of the sentence.

Line 460. Delete "reactive oxygen species" because this acronym is already defined.

Line 466. Delete "abscidic acid" because this acronym is already defined.

Lines 433 and 532-533. Different definitions of the acronyms: Leaves relative water content. In line 433 it is defined as LRWC and in line 533 as RWC. I also recommend defining these acronyms as early as possible in the manuscript.

Lines 431-432, 453. Homogenize the discussion. I recommend always including the common name and the specific name when referring to a plant. In this case, I would suggest indicating that Faba vea is Vicia faba and that Medicago sativa is alfalfa. I recommend reviewing the entire text to locate other cases.

Lines 449-450. The terms chl a, chlb, and Car have already been previously defined.

Lines 532-533. Previously defined acronyms and different acronyms from those indicated in line 433.

Line 563. Delete plant growth-promoting substances such as

Line 651. Correct Sesamun indicum to S. indicum.

Lines 674-675. The authors should review the data presented in these lines, as they suggest that drought treatments are irrigated more frequently than non-stressful conditions. It is recommended that the percentage of irrigation reduction be indicated.

Lines 811-812. The authors should review the abbreviations, as there are many abbreviations presented in the text that are not indicated in this section.

Author Response

Amendments /Responses made according to the reviewers’ recommendations and suggestions

Dear Reviewers,

We sincerely thank you for the time and effort dedicated to reviewing our manuscript. Please rest assured that all the remarks you have provided were carefully considered. The amendments and adjustments made have been highlighted in yellow throughout the manuscript. The table below provides our answers to your comments.

Remarks/ Suggestions of the Reviewer #1 Major Comments

  1. The authors should carefully review the manuscript, especially in the results section. This is because when reading this work, more emphasis is placed on the effect of drought on the plant than on the effect of the biostimulants applied, alone or in combination, in response to drought. Throughout the results, the results obtained from drought are presented first and with greater emphasis, and the data obtained after the application of biostimulants are not compared with the control treatment.

Thank you for this relevant observation. We have revised the Results section to better balance the presentation of drought effects and the impact of the applied biostimulants. More emphasis is now placed on the effects of the individual and combined treatments, and comparisons with the control treatment have been added to highlight the specific contribution of the biostimulants under both well-watered and drought conditions.

  1. Furthermore, in the discussion, more emphasis is placed on the effects of drought on sesame plants than on the effect of the biostimulants, which are truly the object of study. It is particularly striking that in the first part of the discussion (lines 398-505), only the drought results are discussed. It would be advisable to restructure the discussion to give a prominent role to the biostimulants and, especially, to the combined SI+SWE treatment. Furthermore, it would be necessary to compare the results presented with other results related to biostimulants of this type or with biostimulant formulations containing different extracts/compounds of different origins.

We would like to sincerely thank the reviewer for this insightful and constructive comment. In accordance with their recommendation, we have restructured the discussion section. We hope that these adjustments meet the reviewer’s expectations and contribute to improving the scientific clarity and overall quality of the manuscript.

  1. Lines 364-396. The authors should also pay special attention to the correlation results of the DS treatment. Based on what has been explained throughout the work, it is assumed that the DS results are always the same, and that three different trials were not performed. However, it is not clear why the DS results of the correlation are different in the different columns. If the correlation was performed with DS vs. DS+Si, DS vs. DS+SWE, and DS+Si+SWE, this should be clearly indicated in the text.

We thank the reviewer for this pertinent comment and fully agree with the observation regarding the differences in correlation coefficients for the DS treatment. We confirm that the water stress condition (DS) was identical across all treatments and that the same DS data were used in all analyses. However, the correlation analyses were conducted separately, in a comparative manner, each time contrasting the DS treatment with a specific treatment (DS vs DS+Si, DS vs DS+SWE, and DS vs DS+Si+SWE). This approach allows us to evaluate how each treatment distinctly modifies the relationships between physiological parameters under drought stress conditions.

  1. It is recommended, in addition to all the results presented in the manuscript, to also include the results of the impact of the tested biostimulants on sesame production, if available. These data are expected throughout the manuscript since the sesame production cycle is short and it is easy to quantify the g of seeds obtained.

We fully acknowledge the critical importance of yield data (seed mass) for a comprehensive evaluation of the biostimulants’ effects. As detailed in the Methodology section, this study was conducted under controlled conditions over a 63-day period. This duration allowed for detailed analysis of effects on germination, vegetative growth, and initial reproductive stages (including floral initiation), but proved insufficient to observe capsule maturation and seed production stages. This limitation stems from the complete biological cycle of sesame: as demonstrated by scientific literature and our own observations, sesame requires 90-144 days to complete its full phenological cycle (vegetative, reproductive, and maturation phases). Also, we have initiated complementary field trials that will monitor the complete cultivation cycle of sesame. We commit to publishing these yield related findings in future work.

Remarks/ Suggestions of the  Reviewer #1 Minor Comments

  1. Lines 5-6: Add surnames in the same format, preferably in lowercase except for the first letter.

We thank the reviewer for this pertinent observation. As suggested, we have standardized all surnames (lines 5-6) to the recommended format (first letter capitalized, remaining letters in lowercase). For easy verification, these modifications have been implemented in the revised manuscript and are highlighted in yellow as requested.

  1. Line 55: Include sugar between "such as" and "proline."

This modification has been implemented.

  1. Line 66: Introduce the common name.

The recommendation is taken in consideration

  1. Line 74: Replace "which makes" with "making.

The recommendation is taken in consideration

  1. Line 94: Delete "reactive oxygen species," since ROS has already been defined in lines 51-52.

The recommendation is taken in consideration

  1. Line 97: Introduce "reactive water content" (RWC) [8].

This modification has been implemented in the revised manuscript version.

  1. Lines 117-124: Better define the objectives of the work.

We thank the reviewer for this valuable comment. In response, we have revised and clarified the objectives of the study in lines 118 to 130 of the revised manuscript to ensure they are more clearly and explicitly defined.

  1. Line 129: Delete "silicon" and "seaweed," since the acronyms have already been defined in line 83 for silicon, and "SWE" in line 101.

The recommendation is taken in consideration

  1. Line 135. Complement Figure 1 with a boxplot that shows the total plant size, including root and shoot length.

We thank the reviewer for this pertinent suggestion. In order to improve the quality of the manuscript, and in accordance with the recommendation, we have complemented Figure 1 with a boxplot showing the total plant size, including both root and shoot length. This addition allows for a more integrated and comparative visualization of overall plant growth under the different treatments.

  1. Line 136. Correct the figure caption where Figure 2 is indicated, but is Figure 1.

We sincerely thank the reviewer for identifying this numbering error in the figure caption. As suggested, we have corrected this in the revised manuscript.

  1. Line 138. Italicize the specific name Sargassum muticum.

We sincerely thank the reviewer for this valuable observation. This correction has been systematically implemented across all relevant sections.

Remarks/ Suggestions of the  Reviewer #2

  1. In this report the foliar application of two substances, Si and seaweed extracts, called as biostimulants and their combination was evaluated in terms of sesame physiology and morphology under drought stress conditions. Photosystems I and II function, gas exchange, pigment contents and morphological characteristics were measured under well-watered and drought conditions after 64 days treatment in plants grown in a growth chamber. It is well known that Si and SWE can be used in order to alleviate stress symptoms to many plants species. However, this paper is quite interesting, but some changes listed below should be made in order the paper to be accepted.

Keywords: It must be different from the title. Stomatal conductance, quantum yield in PSI and PSII can be used.

We wish to express our sincere gratitude to the reviewer for their thorough evaluation of our work and their highly valuable comments. Their expertise has significantly enhanced the quality of our manuscript. We are honored that they found our study interesting, and we have meticulously implemented all of their recommendations to strengthen both the scientific rigor and clarity of our article.

  1. line 136 correct Fig 1 instead Fig 2.

Thank you for your valuable comment. We appreciate your careful review of our manuscript. Regarding your remark about Line 136, we confirm that the correction ("Figure 1" instead of "Figure 2") has already been addressed in the revised version of the manuscript.

  1. Discussion. Check the first sentence.

Thank you for your helpful comment. Regarding your observation about the first sentence of the Discussion, we confirm that this suggestion has already been incorporated into the revised version of the manuscript.

  1. Conclusions. The two possible strategies of the biostimulants (Si, SWE) should be presented clearly.

We greatly appreciate the reviewer’s valuable comment regarding the clarification of the two potential strategies of the biostimulants (Si and SWE). We would like to highlight that this point has already been addressed and outlined in the Conclusion section of the manuscript.

  1. References. Follow the instructions to the authors, in some cases the format is not relevant.

We have carefully addressed this point in our revised manuscript.

6. I suggest that the manuscript can be accepted for publication after minor revision.

We would like to express our sincere gratitude for your constructive evaluation of our manuscript and for your recommendation for publication after minor revisions. Your positive feedback and encouragement have been truly motivating for our research team. Your comments have been extremely valuable in improving the quality of our work. We particularly appreciate your supportive approach throughout the review process, which has helped us strengthen our manuscript significantly. We have carefully considered each of your suggestions and have incorporated all required modifications in the revised version of the manuscript. The changes made have been highlighted in the document to facilitate your review. We have ensured that the revised version fully meets all of the journal's requirements. Your expertise and advice have been most valuable to us. We thank you once again for both your constructive feedback and your kind encouragement your time and attention have made an important difference in our work.

Remarks/ Suggestions of the  Reviewer #3

  1. Two figures 2, and changes in the text to explain the results.

We sincerely appreciate the reviewer's insightful comment, which has helped strengthen our manuscript.

  1. Capital letters in 2.3 from the results.

We sincerely appreciate the reviewer’s careful observation. The suggested correction has been implemented in the revised version of the manuscript.

  1. From the results, we can see a significant effect of Si and SWE on growth under non-stressed conditions. If the comparison is between controls and treatments, we see improvements in the treated plants; however, these improvements never reach the values of well-watered plants. In the conclusions, it should be emphasised that a 50% reduction in water content is excessive.

Thank you for highlighting this point. We have now added in the Results section an explicit comparison showing that, although Si and SWE improve growth under drought, their values under non stressed conditions still fall short of well-watered plants. This makes clear that a 50 % reduction in soil water content constitutes an extreme stress level, inherently limiting full recovery even with biostimulant application.

  1. One question, I can not see Na+ (Sodium) in the SWE analysis. Why?

We thank the reviewer for raising this important point. We sincerely thank the reviewer for this pertinent remark. In response, we have now added the sodium content (149.11 mg/L) in Table 2, which describes the physicochemical and biochemical composition of the seaweed extract (SWE). We apologize for the initial omission. This value corresponds to approximately 6 mM of Na⁺, a concentration that, based on the literature, does not negatively affect sesame germination or growth. According to Abirami (2023), sesame seeds germinate normally at 50 mM NaCl with 100% germination, although seedling growth is slightly inhibited; inhibition of germination begins at 100 mM and total failure occurs at 250 mM. Additionally, Sridhar et al. (2025) report that shoot and root growth is affected at 100 mM, with nearly complete inhibition at 200 mM. Similar findings by Khademian et al. (2019) confirm growth and yield reduction only at higher salt concentrations (≥80 mM). Therefore, the small amount of sodium introduced by SWE (6 mM) is far below any inhibitory threshold and is unlikely to influence sesame performance negatively during germination or early growth stages.

Abirami, K. Effects of Salinity and Water Stress Factors on Seed Germination, Early Seedling Growth and Proline Content in an Oil Crop, Black Sesame (Sesamum indicum L.). J. Stress Physiol. Biochem. 2023, 19(1), 77–96.

References:

  1. Abirami, K. Effects of salinity and water stress factors on seed germination, early seedling growth and proline content in an oil crop, black sesame (Sesamum indicum L.). J. Stress Physiol. Biochem. 2023, 19(1), 77–96.
  2. Sridhar, D.; Alheswairini, S. S.; Barasarathi, J.; Enshasy, H. A. E.; Lalitha, S.; Mir, S. H.; Sayyed, R. Halophilic Rhizobacteria Promote Growth, Physiology and Salinity Tolerance in Sesamum indicum L. Grown under Salt Stress. Front. Microbiol. 2025, 16, 1590854.
  3. Khademian, R.; Asghari, B.; Sedaghati, B.; Yaghoubian, Y. Plant Beneficial Rhizospheric Microorganisms (PBRMs) Mitigate Deleterious Effects of Salinity in Sesame (Sesamum indicum L.): Physio-Biochemical Properties, Fatty Acids Composition and Secondary Metabolites Content. Ind. Crops Prod. 2019, 136, 129–139.
  4. In Figure 3, the leaf area is represented in two different treatments by two columns with the same letter. However, I cannot believe they are not statistically different.

We sincerely thank the reviewer for this relevant observation. We acknowledge the inconsistency in Figure 3 and have corrected the statistical annotation accordingly to accurately reflect the differences between treatments.

  1. Finally, the conclusion section focuses on the effects of water stress on plants and the beneficial effects of the treatments used. However, less attention has been paid to the impact of the biostimulants on well-watered plants. And the changes in plant growth, photosynthesis, and stomatal conductance, pigments, etc...

We sincerely thank the reviewer for this insightful and valuable comment. We appreciate the careful evaluation of our work. We would like to clarify that this aspect has already been addressed in the Conclusion section.

Reviewer 2 Report

Comments and Suggestions for Authors

In this report the foliar application of two substances, Si and seaweed extracts, called as biostimulants and their combination was evaluated in terms of sesame physiology and morphology under drought stress conditions. Photosystems I and II function, gas exchange, pigment contents and morphological characteristics were measured under well-watered and drought conditions after 64 days treatment in plants grown in a growth chamber. It is well known that Si and SWE can be used in order to alleviate stress symptoms to many plants species. However, this paper is quite interesting, but some changes listed below should be made in order the paper to be accepted.

Keywords: It must be different from the title. Stomatal conductance, quantum yield in PSI and PSII can be used.

  1. line 136 correct Fig 1 instead Fig 2.

Discussion. Check the first sentence.

Conclusions. The two possible strategies of the biostimulants (Si, SWE) should be presented clearly.

References. Follow the instructions to the authors, in some cases the format is not relevant.

I suggest that the manuscript can be accepted for publication after minor revision.

Author Response

Remarks/ Suggestions of the  Reviewer #2

1. In this report the foliar application of two substances, Si and seaweed extracts, called as biostimulants and their combination was evaluated in terms of sesame physiology and morphology under drought stress conditions. Photosystems I and II function, gas exchange, pigment contents and morphological characteristics were measured under well-watered and drought conditions after 64 days treatment in plants grown in a growth chamber. It is well known that Si and SWE can be used in order to alleviate stress symptoms to many plants species. However, this paper is quite interesting, but some changes listed below should be made in order the paper to be accepted.

Keywords: It must be different from the title. Stomatal conductance, quantum yield in PSI and PSII can be used.

We wish to express our sincere gratitude to the reviewer for their thorough evaluation of our work and their highly valuable comments. Their expertise has significantly enhanced the quality of our manuscript. We are honored that they found our study interesting, and we have meticulously implemented all of their recommendations to strengthen both the scientific rigor and clarity of our article.

  1. line 136 correct Fig 1 instead Fig 2.

Thank you for your valuable comment. We appreciate your careful review of our manuscript. Regarding your remark about Line 136, we confirm that the correction ("Figure 1" instead of "Figure 2") has already been addressed in the revised version of the manuscript.

  1. Discussion. Check the first sentence.

Thank you for your helpful comment. Regarding your observation about the first sentence of the Discussion, we confirm that this suggestion has already been incorporated into the revised version of the manuscript.

  1. Conclusions. The two possible strategies of the biostimulants (Si, SWE) should be presented clearly.

We greatly appreciate the reviewer’s valuable comment regarding the clarification of the two potential strategies of the biostimulants (Si and SWE). We would like to highlight that this point has already been addressed and outlined in the Conclusion section of the manuscript.

  1. References. Follow the instructions to the authors, in some cases the format is not relevant.

We have carefully addressed this point in our revised manuscript.

6. I suggest that the manuscript can be accepted for publication after minor revision.

We would like to express our sincere gratitude for your constructive evaluation of our manuscript and for your recommendation for publication after minor revisions. Your positive feedback and encouragement have been truly motivating for our research team. Your comments have been extremely valuable in improving the quality of our work. We particularly appreciate your supportive approach throughout the review process, which has helped us strengthen our manuscript significantly. We have carefully considered each of your suggestions and have incorporated all required modifications in the revised version of the manuscript. The changes made have been highlighted in the document to facilitate your review. We have ensured that the revised version fully meets all of the journal's requirements. Your expertise and advice have been most valuable to us. We thank you once again for both your constructive feedback and your kind encouragement your time and attention have made an important difference in our work.

Reviewer 3 Report

Comments and Suggestions for Authors

Changes must be made in the text and figures:

  1. Two figures 2, and changes in the text to explain the results.
  2. Capital letters in 2.3 from the results.
  3. From the results, we can see a significant effect of Si and SWE on growth under non-stressed conditions. If the comparison is between controls and treatments, we see improvements in the treated plants; however, these improvements never reach the values of well-watered plants. In the conclusions, it should be emphasised that a 50% reduction in water content is excessive.
  4. One question, I can not see Na+ (Sodium) in the SWE analysis. Why?
  5. In Figure 3, the leaf area is represented in two different treatments by two columns with the same letter. However, I can not believe they are not statistically different. 
  6. Finally, the conclusion section focuses on the effects of water stress on plants and the beneficial effects of the treatments used. However, less attention has been paid to the impact of the biostimulants on well-watered plants. And the changes in plant growth, photosynthesis, and stomatal conductance, pigments, etc...

Author Response

Remarks/ Suggestions of the  Reviewer #3

1. Two figures 2, and changes in the text to explain the results.

We sincerely appreciate the reviewer's insightful comment, which has helped strengthen our manuscript.

2. Capital letters in 2.3 from the results.

We sincerely appreciate the reviewer’s careful observation. The suggested correction has been implemented in the revised version of the manuscript.

3. From the results, we can see a significant effect of Si and SWE on growth under non-stressed conditions. If the comparison is between controls and treatments, we see improvements in the treated plants; however, these improvements never reach the values of well-watered plants. In the conclusions, it should be emphasised that a 50% reduction in water content is excessive.

Thank you for highlighting this point. We have now added in the Results section an explicit comparison showing that, although Si and SWE improve growth under drought, their values under non stressed conditions still fall short of well-watered plants. This makes clear that a 50 % reduction in soil water content constitutes an extreme stress level, inherently limiting full recovery even with biostimulant application.

4. One question, I can not see Na+ (Sodium) in the SWE analysis. Why?

We thank the reviewer for raising this important point. We sincerely thank the reviewer for this pertinent remark. In response, we have now added the sodium content (149.11 mg/L) in Table 2, which describes the physicochemical and biochemical composition of the seaweed extract (SWE). We apologize for the initial omission. This value corresponds to approximately 6 mM of Na⁺, a concentration that, based on the literature, does not negatively affect sesame germination or growth. According to Abirami (2023), sesame seeds germinate normally at 50 mM NaCl with 100% germination, although seedling growth is slightly inhibited; inhibition of germination begins at 100 mM and total failure occurs at 250 mM. Additionally, Sridhar et al. (2025) report that shoot and root growth is affected at 100 mM, with nearly complete inhibition at 200 mM. Similar findings by Khademian et al. (2019) confirm growth and yield reduction only at higher salt concentrations (≥80 mM). Therefore, the small amount of sodium introduced by SWE (6 mM) is far below any inhibitory threshold and is unlikely to influence sesame performance negatively during germination or early growth stages.

Abirami, K. Effects of Salinity and Water Stress Factors on Seed Germination, Early Seedling Growth and Proline Content in an Oil Crop, Black Sesame (Sesamum indicum L.). J. Stress Physiol. Biochem. 2023, 19(1), 77–96.

References:

1.      Abirami, K. Effects of salinity and water stress factors on seed germination, early seedling growth and proline content in an oil crop, black sesame (Sesamum indicum L.). J. Stress Physiol. Biochem. 2023, 19(1), 77–96.

2.     Sridhar, D.; Alheswairini, S. S.; Barasarathi, J.; Enshasy, H. A. E.; Lalitha, S.; Mir, S. H.; Sayyed, R. Halophilic Rhizobacteria Promote Growth, Physiology and Salinity Tolerance in Sesamum indicum L. Grown under Salt Stress. Front. Microbiol. 2025, 16, 1590854.

3.     Khademian, R.; Asghari, B.; Sedaghati, B.; Yaghoubian, Y. Plant Beneficial Rhizospheric Microorganisms (PBRMs) Mitigate Deleterious Effects of Salinity in Sesame (Sesamum indicum L.): Physio-Biochemical Properties, Fatty Acids Composition and Secondary Metabolites Content. Ind. Crops Prod. 2019, 136, 129–139.

5. In Figure 3, the leaf area is represented in two different treatments by two columns with the same letter. However, I cannot believe they are not statistically different.

We sincerely thank the reviewer for this relevant observation. We acknowledge the inconsistency in Figure 3 and have corrected the statistical annotation accordingly to accurately reflect the differences between treatments.

6. Finally, the conclusion section focuses on the effects of water stress on plants and the beneficial effects of the treatments used. However, less attention has been paid to the impact of the biostimulants on well-watered plants. And the changes in plant growth, photosynthesis, and stomatal conductance, pigments, etc...

We sincerely thank the reviewer for this insightful and valuable comment. We appreciate the careful evaluation of our work. We would like to clarify that this aspect has already been addressed in the Conclusion section.

Round 2

Reviewer 3 Report

Comments and Suggestions for Authors

Accept in the present form.